

# Characterising low-cost sensors in highly portable platforms to quantify personal exposure in diverse environments.

Lia Chatzidiakou[1], Anika Krause[1], Olalekan A.M. Popoola[1], Andrea Di Antonio[1], Mike Kellaway[2], Yiqun Han[3,4,5], Freya A. Squires[6], Teng Wang[4,7], Hanbin Zhang[3,5,8], Qi Wang[4,7], Yunfei Fan[4], Shiyi Chen[4], Min Hu[4,7], Jennifer K. Quint[9], Benjamin Barratt[3,5,8], Frank J. Kelly[3,5,8], Tong Zhu[4,7], Roderic L. Jones[1]

[1]Department of Chemistry, University of Cambridge, Cambridge, CB2 1EW, UK
[2]Atmospheric Sensors Ltd, Bedfordshire, SG19 3SH, UK
[3] MRC-PHE Centre for Environment & Health, Imperial College London and King's College London, London, W2 1PG, UK
[4] College of Environmental Sciences and Engineering, Peking University, Beijing, 100871, China
[5] Department of Analytical, Environmental and Forensic Sciences, King's College London, London, SE1 9NH, UK
[6] Department of Chemistry, University of York, York, YO10 5DD, UK
[7]The Beijing Innovation Center for Engineering Science and Advanced Technology, Peking University, Beijing, 100871, China.
[8] NIHR Health Protection Research Unit in Health Impact of Environmental Hazards, King's College London, London, SE1 9NH, UK
[9] National Heart and Lung Institute, Imperial College London, SW3 6LR, UK

*Correspondence to*: Dr Lia Chatzidiakou, email: ec571@cam.ac.uk, Tel: 01223 336345

**Abstract**. The inaccurate quantification of personal exposure to air pollution introduces error and bias in health estimations, severely limiting causal inference in epidemiological research worldwide. Rapid advancements in affordable, miniaturised air pollution sensor technologies offer the potential to address this limitation by capturing the high variability of personal exposure during daily life in large-scale studies with unprecedented spatial and temporal resolution. However, concerns remain regarding the suitability of novel sensing technologies for scientific and policy purposes. In this paper we characterise the performance of a portable personal air quality monitor (PAM) that integrates multiple miniaturised sensors for nitrogen oxides ($NO_x$), carbon monoxide (CO), ozone ($O_3$) and particulate matter (PM) measurements along with temperature, relative humidity, acceleration, noise and GPS sensors. Overall, the air pollution sensors showed excellent agreement with standard instrumentation in outdoor, indoor and commuting microenvironments across seasons and different geographical settings. An important outcome of this study is that the error of the PAM is significantly smaller than the error introduced when estimating personal exposure based on sparsely distributed outdoor fixed monitoring stations. Hence, novel sensing technologies as the ones demonstrated here can revolutionise health studies by providing highly resolved reliable exposure metrics at large scale to investigate the underlying mechanisms of the effects of air pollution on health.

**Keywords.** personal exposure, portable air quality monitor, miniaturised sensor technologies, nitrogen oxides ($NO_x$), carbon monoxide (CO), particulate matter (PM)

## 1 Introduction

Emerging epidemiological evidence has associated exposure to air pollution with adverse effects on every major organ system (Thurston et al., 2017). Most of this evidence comes from western Europe and North America (Newell, Kartsonaki, Lam, & Kurmi, 2017) as population-scale air pollution health studies have largely relied on available outdoor air pollution measurements from fixed monitoring stations (COMEAP, 2018). Due to limitations in the availability of monitoring networks in low- and middle-income countries (LMICs), the effects of air pollution on health have been under-researched in these settings. A clear need exists for more direct epidemiological evidence in diverse geographical settings with varying air pollution sources considering the high likelihood that health effects of air pollution are not linear, and cannot be simply transcribed from the western world to LMICs (Tonne, 2017).

Secondly, the low spatial and temporal resolution of exposure metrics at postcode level or coarser which are often employed in large-scale epidemiological research cannot separate the individual health effects of pollutants which are generally highly correlated at these coarser scales. Additionally, outdoor measurements cannot capture the *total* personal exposure that results from the cumulative effects of an individual moving between different indoor and outdoor microenvironments. During daily



life, peak exposure events often occur during commuting (Karanasiou, Viana, Querol, Moreno, & de Leeuw, 2014) while the indoor environment is a significant site for exposure in part because people spend as much as 90% of their time indoors (Klepeis et al., 2001). Indoor air is affected by outdoor pollutants penetrating building envelopes with additional indoor sinks, sources and emissions from building materials which cannot be detected by fixed outdoor monitoring networks. The lack of

information on indoor environments at population scale is a significant factor in poorly quantified health risks. As a result, inaccurate personal exposure estimations to air pollution introduce both bias and error in health estimations ultimately preventing epidemiological research to move from general associations to the specific (Zeger et al., 2000).

Rapid advancements in novel sensing technologies of air pollution sensors now offer the potential to monitor detailed personal
exposure during daily life at population scale, thanks to their significantly reduced cost, smaller size and fast-response. Instrument development is accelerating fast with a growing number of companies utilising combinations of such sensors (Cross et al., 2017) as well as auxiliary components to build different types of monitors (Morawska et al., 2018). As a special case, it is now estimated that there are currently over 30,000 sensors operating in China to monitor concentrations of air pollutants (Morawska et al., 2018). Several studies over the last 15 years have attempted to quantify personal exposure to air pollutants
by employing portable sensors, but most of those studies have been restricted to small-scale surveys (Steinle, Reis, & Sabel, 2013). However, large-scale studies are necessary to assess the health effects of harmful pollutants because they are often seen in only small subgroups of the population due to varying individual susceptibility and exposure profiles. Novel sensing technologies are in fact the only method to expand the personal exposure coverage at the population level. Yet, concerns remain about the validation and quality control of those sensors (Castell et al., 2017) as few personal exposure studies have
evaluated their performance in field deployment conditions (Rai et al., 2017). Typically, novel sensing platforms are exclusively evaluated in outdoor static co-locations with reference instruments and they only target small numbers of pollutants, most commonly ozone, nitrogen dioxide (Lin et al., 2015) and/or particulate matter (Holstius, Pillarisetti, Smith, & Seto, 2014)(Feinberg et al., 2018).

To address these shortcomings, a highly portable personal air pollution monitor (PAM) that measures a large number of chemical and physical parameters simultaneously has been developed. This paper aims to evaluate the performance of the PAM when capturing *total* personal exposure to air pollution in diverse environmental conditions. To do so, the PAM performance was assessed in well-characterised outdoor, indoor and commuting microenvironments across seasons and different geographical settings. The PAM has already been deployed to participants of two large cardio-pulmonary cohorts in
China (Han et al., 2019) (AIRLESS- Theme 3 APHH project) (Shi et al., 2018) and the UK (COPE) (Moore et al., 2016), and in a number of smaller international pilot projects in Northern America, Europe, South and East Asia and Africa. This is the first of a series of publications that aim to capture *total* personal exposure to a large number of pollutants at unprecedented detail, and together with medical outcomes, to identify underlying mechanisms of specific air pollutants on health. As the field of novel air pollution sensing technologies expands rapidly, this paper further aims to create a roadmap for calibration and
validation of portable monitors suitable for personal exposure quantification.

## 2 The personal air quality monitor

The PAM (Figure 1) is an autonomous platform that incorporates multiple sensors of physical and chemical parameters (Table 1). The compact and lightweight design of the PAM (ca. 400g) makes the unit suitable for personal exposure assessment. The PAM is almost completely silent and can operate continuously. No other input is required by the user other than to place it for
periodic charging (e.g. daily) and data upload in a base station. The measurements are also stored in an SD card inside the monitor and uploaded through general packet radio service (GPRS) to a secure access FTP server. Customised system software



has been developed to optimise the performance of the platform. Depending on the chosen sampling interval of either 20 sec or 1 min, the battery life on a single charge lasts for 10 hours or 20 hours respectively. The combined cost of the sensors alone is less than £600 and the total cost of the PAM is less than £2,000 making it a "lower-cost" system (Cross et al., 2017).

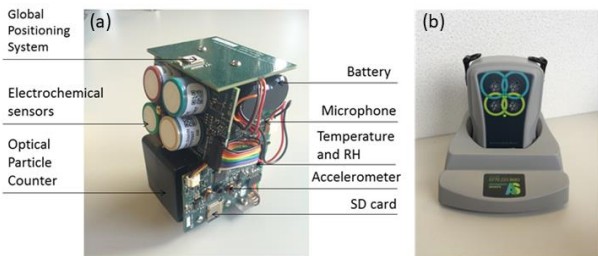

**Figure 1. The personal air quality monitor. (a): design of the PAM platform internals and (b): in charging base-station. The external dimensions of the PAM are 13cm * 9 cm * 10 cm.**

**Table 1. Summary of monitored parameters of the PAM. $PM_1$, $PM_{2.5}$ and $PM_{10}$ = the fraction of particles with an aerodynamic diameter smaller than 1 μm, 2.5 μm and 10 μm respectively; CO = carbon monoxide; NO = nitric oxide; $NO_2$ = nitrogen dioxide; $O_3$**
**= ozone.**

| Parameter | Method | Sampling Interval |
|---|---|---|
| Spatial coordinates | Global Positioning System (GPS) | 20 sec |
| Background noise | Microphone | 100 Hz |
| Physical activity | Tri-axial accelerometer | 100 Hz |
| Temperature | Band-gap IC | 4 sec |
| Relative Humidity (RH) | Capacitive | 4 sec |
| $PM_1$, $PM_{2.5}$, $PM_{10}$ | Optical Particle Counter (OPC) | 20 sec |
| CO, NO, $NO_2$, $O_3$ | Electrochemical sensors (EC) | 100 Hz |

User-friendly, bespoke software (Supplementary material Figure A1) has been developed to automate the management and post-processing of the large volume of raw data collected with the PAM network. Data is held in a PostgreSQL relational database management system which has an unlimited row-storage capacity and allows the querying of large quantities of data in a flexible manner while maintaining performance as the volume of data grows. Post-processing was performed in R software
(R Development Core Team, 2008) (Supplementary material Figure A1) following the methodology outlined in this paper.

### 2.1 Measurements of CO, NO, $NO_2$ and $O_3$

The principle of operation of all commercially available miniaturised gaseous sensors currently involves measuring changes in specific properties of a sensing material (e.g. electrical conductivity, capacitance, mass, optical absorption) when exposed
to a gas species (Morawska et al., 2018). The PAM integrates small (20mm diameter) electrochemical (EC) sensors based on an amperometric principle of operation (Stetter & Li, 2008) for the quantification of carbon monoxide (CO), nitric oxide (NO), nitrogen dioxide ($NO_2$) and ozone ($O_3$). These EC sensors are the A4 variant from Alphasense (NO-A4(Alphasense Ltd, 2016a), CO-A4 (Alphasense Ltd, 2017a), NO2-A43F (Alphasense Ltd, 2016b), Ox-A431 (Alphasense Ltd, 2017b)) and operate on a four electrode system, where the conventional setup of working electrode, counter electrode and reference
electrode is supplemented with an additional electrode, the auxiliary (or non-sensing) electrode to compensate for the temperature dependence of the cell potential (Popoola, Stewart, Mead, & Jones, 2016). Earlier variants of EC sensors used in this paper have been extensively characterised in laboratory conditions and in static outdoor dense sensor networks (Mead et al., 2013). Those studies provided evidence that, after appropriate post-processing, the sensors had a linear response to the





targeted pollutants and achieved excellent performance with limits of detection (LOD) < 4ppb demonstrating their suitability for atmospheric air quality measurements.

Currently, standards for the calibration and performance evaluation of EC sensors focus on industrial applications (British Standards Institution, 2017). Following those standards, a widely adopted approach to calibrate EC sensors are gas chamber experiments to determine offset (baseline) and sensitivity (gain). To address the lack of standards for novel sensing technologies, a number of researchers and governmental organisations are developing protocols and guidelines to evaluate sensor/monitor performance in the laboratory and in the field, such as the European Metrology Research Programme of EURAMET (Spinelle, Aleixandre, Gerboles, & European Commission. Joint Research Centre. Institute for Environment and Sustainability., 2013), the European Standardisation Committee (CEN/TC 264/WG 42, 2018) and US-based groups (Long, Beaver, & Williams, 2014) (AQ-Spec, 2017).

Building on those protocols, the EC sensors were calibrated by co-location with certified reference instruments in similar environmental conditions and same geographical area where the monitors had been or were to be deployed. The considerable advantage of this approach over laboratory calibration includes the exposure of the sensor to the actual air pollution and temperature/relative humidity conditions under which it is expected to operate, as well as the assessment of any site-specific potential cross-interferences. A linear regression model (Equation 1) was applied to the co-location data to determine the calibration parameters used to convert raw sensor signals (mV) to mixing ratios (ppb). Temperature effects were corrected through the auxiliary electrode AE which might have a different sensitivity to the working electrode WE ($a \neq b$). The cross-sensitivities between the $NO_2$ and $O_3$ measurements were corrected via parameter $c$ (the cross sensitive gas $Y$ is $NO_2$ for $O_3$ measurements and vice versa). As the CO and NO sensors were found to be sufficiently selective, $c$ was set to zero for the calibration of those sensors.

$$[\mathbf{X}]_{\mathbf{ref}} = \mathbf{a}\ \mathbf{WE_X} + \mathbf{b}\ \mathbf{AE_X} + \mathbf{c}\ \mathbf{WE_Y} + \mathbf{d} \qquad \textbf{Equation 1}$$

*where*

| | |
|---|---|
| $[X]_{ref}$ | reference measurement of pollutant X [ppb] |
| $a$ | sensitivity of the working electrode [ppb / mV] |
| $WE_X$ , $AE_X$ | raw signal of the working and auxiliary electrode [mV] |
| $b$ | sensitivity of the auxiliary electrode [ppb / mV] (accounts for temperature) |
| $c$ | cross sensitivity with gas Y [ppb / mV], $c = 0$ for CO and NO |
| $WE_Y$ | raw signal of the working electrode of the cross sensitive gas Y [mV] |
| $d$ | intercept [ppb] |

To evaluate the performance of the linear model, the datasets were split into training (i.e. calibration) and validation periods to first extract the calibration parameters and then apply them to the validation set and compare the measurements with those from reference instruments (referred to as "calibration - validation" method). As relationships in these models should ideally not be extrapolated beyond the range of the observations (including meteorological conditions), the calibration periods were made sufficiently long to cover the temperature and concentration ranges in which the sensors were deployed (Cross et al., 2017).

## 2.2 Particulate mass measurements

The operation of virtually all miniaturised particulate matter (PM) sensors that are currently commercially available is based on the light scattering principle, either volume scattering devices or optical particle counters (OPCs) (Morawska et al., 2018). The PAM integrates a commercially available miniaturised OPC (Alphasense OPC-N2) (Alphasense Ltd, 2018) which uses



Mie scattering for real-time aerosol characterisation. Particles pass through a sampling volume illuminated by a light source (in this case a laser) and scatter light into a photo detector. The amplitudes of the detected scattering signals pulses are then related to particle size. The OPC counts these pulses and typically sorts them into different particle size bins (Walser, Sauer, Spanu, Gasteiger, & Weinzierl, 2017). The OPC-N2 classifies particles in 16 sizes (bins) in the range 0.38 -17 μm. The

performance of this OPC in the laboratory (AQ-Spec, 2017) (Sousan, Koehler, Hallett, & Peters, 2016) showed a high degree of linearity. Similarly studies evaluating the OPC performance in outdoor static deployments (Di Antonio et al., 2018) (Crilley et al., 2018) showed that once site- and seasonally-specific calibrations were applied, the miniaturised sensor could be used to quantify number and mass concentrations of particles with a precision similar to other standard commercial reference optical PM instruments.

The complexity of evaluating PM sensor performance is much greater than that of gas sensors. Compared with standard instrumentation, optical PM instruments face four inherent limitations which introduce potential differences in mass estimations compared with reference gravimetric methods:

    (a) Exposure of the particles to relative humidity (RH) results in hygroscopic growth of particles and leads to mass
15           overestimation (Di Antonio et al., 2018).

    (b) Small variations in the sensitivities of the photodetector and the intensity/ angle of the laser may result in a systematic error specific to each OPC sensor. Additionally, as particles enter the optical chamber, they may deposit on internal surfaces and optics of the sensor leading to a reduction in the measured scattered light, and thus instrument sensitivity.

    (c) A further limitation of all optical methods is their inability to detect particles with a diameters below a certain size,
20           typically 200-400 nm (Morawska & Salthammer, 2003).

    (d) Finally, optical methods cannot distinguish the physical and chemical parameters of the aerosol (e.g. density, hygroscopicity, volatility) which might vary significantly as people move between different microenvironments with diverse emission sources further increasing the uncertainty of mass estimation.

To compensate for these limitations, this work firstly corrected for the effect of RH by applying an algorithm based on the particle size distribution which was developed for aerosols in urban environments (Di Antonio et al., 2018) (a). In the second step, a scaling factor for each OPC was determined to account for sensor-sensor variability (b). This scaling factor was determined from a linear fit between the RH-corrected mass and the reference measurements. As the reference instruments (e.g. TEOM) include particles below the size range of the OPC in their mass estimations, the scaling factor addresses partly
the under-prediction of mass due to undetected smaller particles (c). The varying aerosol composition (d) remains a challenge, and therefore a constant density of 1.65 g/cm$^3$ was assumed. Although the OPC is able to measure PM$_1$, PM$_{2.5}$ and PM$_{10}$, this paper focusses on the performance of the PM$_{2.5}$ measurements because of the availability of reference instruments.

**3 Performance of the PAM under well-characterised conditions in the field**

In the following three sections the performance of the PAM is assessed when measuring air pollution concentrations in different
environments that are relevant for the quantification of *total* personal exposure (outdoor, indoor and in movement) in the UK and China. Sensor performance may vary significantly with season (e.g. temperature and RH artefacts) while meteorological conditions may affect the variation of outdoor air pollution levels directly (e.g. stability of the atmosphere) and indirectly by socio-economic patterns (e.g. increased energy demand for heating). Similarly, indoor air may be directly affected by outdoor air pollution levels and indirectly through occupants' behavioural patterns (e.g. windows' adjustment to achieve thermal
comfort). Taking into account the strong seasonal variation of air pollution levels, the performance of the PAM was evaluated by co-locating one or multiple PAMs with reference instruments both during the "heating" (when the majority of householders





heat their home on a regular basis) and "non-heating" season. The residential central heating season in Beijing is from 15th November – 15th March (Beijing Municipal Government), while in the UK the equivalent heating season is 5.6 months (October – March/April) (BRE, 2013).

5   The description of the sites, principle of operation and models of certified reference instrumentation used can be found in Table 2. The co-locations in China involved 60 PAMs which had been previously deployed to 250 participants of a cardio-pulmonary cohort for one month during the heating and one month during the non-heating season (Han et al., 2019). The co-location in the UK involved 60 PAMs that have been previously deployed to 150 participants of a COPD cohort for two years continuously (Moore et al., 2016). The reproducibility between co-located sensors was very high even when the ambient concentrations

10  were close to the LOD (mean $R^2 \geq 0.80$ for EC sensors and $R^2 \geq 0.91$ for the OPC, see Figure 2 and Supplementary material Table T1). Hence, the performance of the selected PAMs in static deployments as described in this section is representative and can be extrapolated to the entire sensor network.

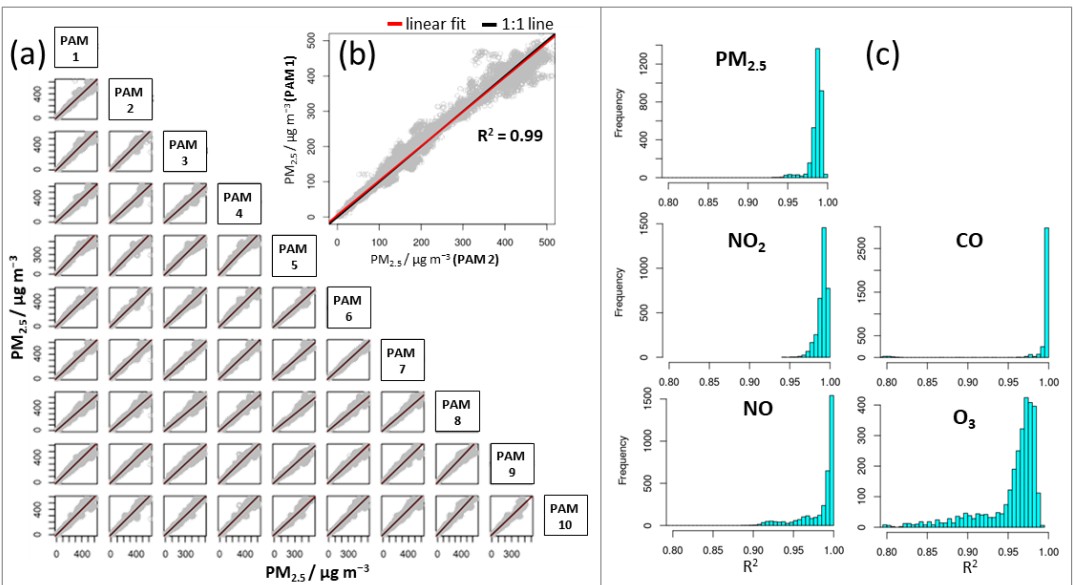

**Figure 2. Reproducibility of a PAM network (in that case 60) co-located outdoors in Beijing during the heating season after one month of field deployment. (a): Scatterplot of the PM$_{2.5}$ measurements between 10 sensor pairs. The 1:1 line in black; and linear fit line in red. (b): "Close-up" of a scatterplot from (a) of one representative sensor pair. (c): histogram of the coefficient of determination ($R^2$) between all sensor pairs. $R^2$ values during this deployment were higher than 0.90 for all pollutants indicating the high reproducibility of the sensors' readings (see Table T1 for all co-locations). O$_3$ sensors $R^2 > 0.80$ due to very low ambient levels close to the LOD of the sensors.**





**Table 2. Details of the reference instruments used in this study. Time resolution of all measurements was 1 minute.**
**\* Due to malfunctioning of the TEOM in PKU during non-heating season, measurements from a TEOM in a nearby governmental site (Haidianwanliu, time resolution 1 hour) were used.**

| Deployment | Site description | NO, NO$_2$ | CO | PM | O$_3$ |
|---|---|---|---|---|---|
| **Outdoor China** | Urban background in Peking University (PKU) campus, Beijing | Chemiluminesence, Thermo-Science Model 42i | Nondispersive Infrared, Thermo-Science Model 48i | PM$_{2.5}$ * TEOM (Tapered Element Oscillating Microbalance) | UV absorption Thermo Model 49i |
| **Outdoor UK** | Urban background at the Department of Chemistry, Cambridge | Chemiluminesence, Thermo-Science Model 42i | Nondispersive Infrared, Thermo-Science Model 48i | aerosol spectrometer FIDAS PALAS 200S | UV absorption Thermo-science Model 49i |
| **Indoor residential China** | Indoor deployment in an urban high-rise Beijing flat | NO$_2$ cavity attenuated phase shift spectroscopy (CAPS) Teledyne API T500U | NA | aerosol spectrometer GRIMM 1.108 | NA |
| **Commuting environment UK** | Monitoring Vehicle equipped with commercial instruments driving in central London | NO$_2$ CAPS Teledyne API T500U | NA | Nephelometer (scattering) Met One ES642 | UV absorption Teledyne API T400 |

**3.1 Outdoor performance of sensors in diverse urban environments with varying pollution profiles and meteorological parameters**

In total, four outdoor co-location deployments have been evaluated to comprehensively characterise the performance of the sensors (in the UK and China during the heating and non-heating season, see Table 3). The PAMs were placed in protective shelters close to the inlets of the certified air pollution monitoring stations. The sensor measurements were converted to physical units following the methodology described in subsections 2.1 and 2.2.

As an illustrative example, the outdoor co-location in Beijing, China (19 days, Dec 2016 to Jan 2017) is presented in Figure 3 to demonstrate the previously mentioned "calibration - validation" method (subsection 2.1). The time series of the pollutants measured by the PAM (blue) follow closely the reference instruments (red) in both the calibration (Figure 3a) and validation (Figure 3b) periods. Similarly, the time series and scatterplots of the other three co-locations (UK in the heating season, China and UK in the non-heating season) can be found in the Supplementary material (Figures A2, A3, A4).





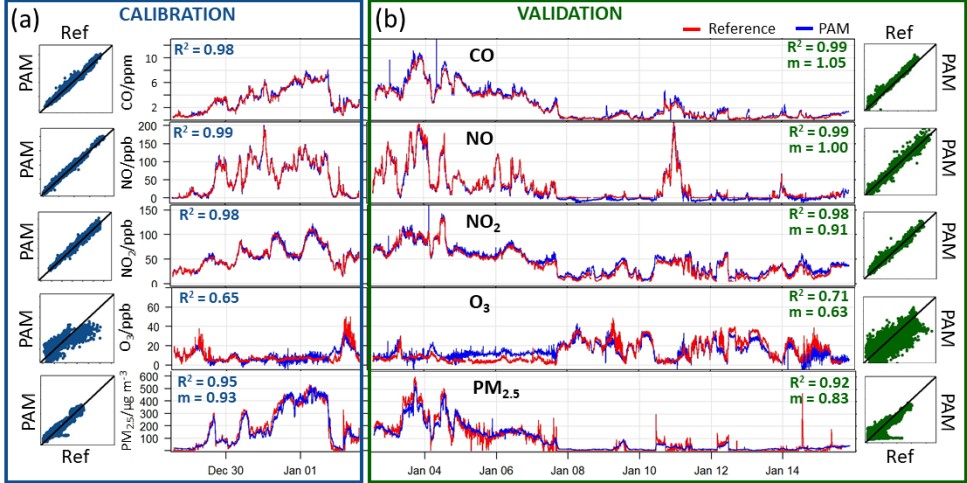

**Figure 3.** Outdoor co-location of one representative PAM with calibrated reference instruments in China (winter 2016/17) at 1-min time resolution demonstrating the "calibration-validation" methodology to evaluate the performance of the linear model. The first five days (a) were used to calibrate the EC sensors. The remaining co-location data (14 days, b) were used to validate the extracted calibration parameters. The scatterplots on each side show the correlations between reference and PAM measurements with the 1:1 line in black. $\overline{R^2}$ and gradients (m) are shown on each side in the corresponding colour.

Table 3 gives a quantitative overview of the agreement between the PAM measurements and the reference instruments in outdoor co-locations during the heating and non-heating season. Ambient temperature and RH (median, range: 5% -95%) as well as the maximum pollutant concentration measured are presented to describe the ambient conditions of each co-location. Because the PAM internal temperature is on average 7°C higher than the ambient temperature due to heat generated by the internal battery, the internal conditions the sensors were exposed to are also presented. The sensor performance against the reference instruments was evaluated using (1) the adjusted coefficient of determination ($\overline{R^2}$) of the linear regression between PAM and reference and (2) the root mean square error (RMSE) using both the validation and calibration period (Table 3). $\overline{R^2}$ may be a misleading indicator of sensor performance when measurements are taken close to the LOD of the instruments. The RMSE can be a complementary parameter of $\overline{R^2}$ for the evaluation of performance, as it summarises the mean difference between measurements from the sensor and certified instruments The average values of $\overline{R^2}$ and RMSE of all N sensors during all co-locations are given in Table 3.





Table 3. Overview of sensors' performance during outdoor co-locations in China and the UK (7 to 19 days). Median values (range: 5th - 95th percentile) of the ambient temperature and relative humidity (RH), internal temperature and RH of the platform are presented. The 95th percentile of the concentration measurements of the reference over the entire co-location period is given as maximum concentration for each pollutant. The mean adjusted coefficients ($\overline{R}^2$) and root mean square errors (RMSE) indicate the agreement between the measurements of the sensors and reference instruments. The average values of all N sensors for each variable are given. Co-location in China in June shown in Italics as sensors were regularly exposed to temperatures higher than 40°C where sensors do not show linear temperature responses. The sensor reproducibility for these co-locations is presented in Table T1 in the Supplementary material.

| | | Heating season | | Non-heating season | |
|---|---|---|---|---|---|
| | | **China (Dec- Jan)** | **UK (Oct- Nov)** | *China (June)* | **UK (April-May)** |
| **Illustrative graphical example** | | Figure 3 | Figure A2 | *Figure A3* | Figure A4 |
| **Ambient conditions** | Ambient Temp (°C) | 1.1 (-3.6 – 6.1) | 9.3 (4.3-14.4) | *29.9 (22.8-36.3)* | 9.5 (4.7-18.1) |
| | Ambient RH (%) | 40 (15-79) | 81 (61-93) | *68 (43-96)* | 82 (48-93) |
| **Internal conditions** | Internal Temp (°C) | 10.5 (5.3-18.0) | 15.9 (11.0–20.8) | *40.2 (32.7–45.8)* | 17.7 (12.2–26.8) |
| | Internal RH (%) | 27 (14-44) | 52 (39 -59) | *38 (23 – 55)* | 52 (34 – 60) |
| **Number of sensors (N)** | [-] | N =59 | N=3 | *N=59* | N= 3 |
| **CO** | Maximum mixing ratio (ppb) | 6845 | 357 | *916* | 276 |
| | $\overline{R}^2$ | 0.98 | 0.74 | *0.71* | 0.67 |
| | RMSE in ppb (percentage of max) | 31 (0.5%) | 31.6 (8.9%) | *212 (23%)* | 33.3 (12.1%) |
| **NO** | Maximum mixing ratio (ppb) | 132 | 19 | *5* | 6 |
| | $\overline{R}^2$ | 0.94 | 0.89 | *0.20* | 0.58 |
| | RMSE in ppb (percentage of max) | 11.7 (8.9%) | 3.0 (15.8%) | *13.0 (260%)* | 2.2 (36.6%) |
| **NO₂** | Maximum mixing ratio (ppb) | 98 | 35 | *42* | 19 |
| | $\overline{R}^2$ | 0.84 | 0.90 | *0.20* | 0.84 |
| | RMSE in ppb (percentage of max) | 11.8 (12.0%) | 3.0 (8.6%) | *13.3 (31.7%)* | 2.6 (13.7%) |
| **O₃** | Maximum mixing ratio (ppb) | 33 | 30 | *109* | 44 |
| | $\overline{R}^2$ | 0.87 | 0.92 | *0.80* | 0.89 |
| | RMSE in ppb (percentage of max) | 3.6 (10.9%) | 2.7 (9%) | *14.9 (13.7%)* | 4.2 (9.5%) |
| **PM₂.₅** | Maximum conc. (µg m⁻³) | 432 | 32 | *110* | 37 |
| | $\overline{R}^2$ | 0.93 | 0.57 * | *0.65 \*\** | 0.80 |
| | RMSE in µg m⁻³ (percentage of max) | 37 (8.6%) | 9 (28%) * | *25 (22.7%)\*\** | 2 (5.4%) |

\* due to unavailable data, PM mass measurements are not corrected for RH effects
\*\*comparison with governmental station ~ 3km away

### 3.1.1 Outdoor performance of the PAM during the heating season co-locations

During the heating season outdoor co-locations of a number of PAMs next to certified reference instruments, ambient temperatures ranged from –4°C to 6°C in China and between 4°C and 14°C in the UK. Air pollution in China was characterised by elevated levels of CO and PM₂.₅ (Table 3) for extended time periods ("*haze*" events) partially driven by stagnant winds or a weak southerly wind circulation (Shi et al., 2018). Compared with pollutant





levels in the UK, the concentrations of these CO and PM$_{2.5}$ were approximately ten times higher while the contrast in ambient NO$_2$ levels was less marked with levels in China only approximately three-fold higher.

The O$_3$, NO and NO$_2$ sensors exhibited an excellent performance ($\overline{R^2} \geq 0.84$) in both geographical settings (Table 3). The median RMSE values were close to the LOD of the sensors (< 3 ppb) in the UK and slightly higher in China (< 12 ppb) (Figure 3, Table 3). In both deployments, the RMSE values of these gaseous sensors were negligible compared to the ambient concentration ranges of the targeted pollutants (less than 16% of the maximum mixing ratio recorded by the reference instruments). While the median $\overline{R^2}$ between the CO sensor and the corresponding reference was reasonably high in both outdoor deployments ($\geq 0.74$), the median RMSE values were also quite large (< 32 ppb). In fact, this is due to the known high intrinsic noise and LOD of the reference instrumentation (> 40 ppb) which is much higher compared to that of the electrochemical sensors (LOD < 4 ppb, see subsection 2.1).

Following the correction of the size segregated particle measurements for the effect of RH (subsection 2.2) the PM mass quantification with the miniaturised OPC agrees with the TEOM reference instrument with an adjusted $\overline{R^2}$ of 0.93. The low RMSE values (> 8.6% of the maximum concentration) demonstrate that the scaling factor addresses adequately the under-prediction of mass due to undetected smaller particles when derived from field calibration in the local environment. Due to unavailable measurements, the PM measurements in the UK could not be corrected for RH effects which resulted in only a moderate correlation with the reference instrument ($\overline{R^2}$ = 0.57, Figure A2).

### 3.1.2 Outdoor performance of the PAM during the non-heating season co-locations

One outdoor co-location in China (Figure A3) and one in the UK (Figure A4) were performed during the non-heating season, both over periods of two weeks (Table 3). In the UK, seasonal variation of ambient temperatures, RH and pollution levels was relatively small. In contrast, in China, seasonal variation was large with ambient temperatures reaching up to 36.3°C (median: 29.9°C) and generally lower pollution levels compared to the heating season. However, in both geographical settings, O$_3$ was significantly elevated. The performance of the O$_3$ sensor remained reliable in all deployments with median $\overline{R^2}$ = 0.80 and RMSE values <15 ppb, which might provide valuable insights on the health effects of this pollutant because (a) ozone is a strong oxidant with a high potential to affect the body and (b) has highest concentrations during the non-heating season compared to other pollutants which usually peak during the heating season.

Due to a malfunction of the PM reference (TEOM) instrument during the non-heating season at PKU, the PAM PM measurements had to be compared with a TEOM installed in a nearby governmental site ("*Haidianwanliu*"). Although not closely co-located (~ 3km), the gradient between the PAMs and reference measurements was close to unity (average m = 0.96, see example Figure A3) and there was still a notable correlation ($\overline{R^2}$ = 0.65) with a median RMSE of 25 μg m$^{-3}$ indicating that away from direct sources, PM concentrations are essentially homogenous over relatively large urban areas. Compared with the heating season, PM concentrations in China were significantly lower, whereas PM levels in the UK varied little with season. After correcting for the effects





of RH on PM, the PAM performance in the UK during the non-heating season significantly improved compared with the heating season (RMSE = 2 μg m⁻³ within the particle size range 0.38 – 17 μm).

While the performance of the $O_3$ and OPC sensor remained reliable across seasons and geographical settings, the
performance of the CO, NO and $NO_2$ sensors decreased significantly ($\overline{R^2} \geq 0.20$) during the non-heating season in China due to extreme temperatures (median: 40.2°C, 5-95%: 32.7°C– 45.8°C, Table 3). It should be noted that NO levels were close to the LOD of the sensor which also affects the $\overline{R^2}$ values. We conclude that the measurements of the CO, NO and $NO_2$ sensors should be interpreted with caution when the PAM is exposed to temperatures above 40°C. However, during the field deployment to participants, the sensors were exposed to lower
temperatures (see Figure A5) that did not impact on their performance (see subsection 3.2).

### 3.2 Indoor performance of the $NO_2$ and PM sensors

Low-cost air pollution sensors have generally been characterised outdoors next to reference instruments as described in the previous section. However, little is known about the performance of these sensors in indoor environments, where the population spends most of their time (Klepeis et al., 2001), where environmental
conditions (e.g. temperature, RH) and emission sources may be significantly different compared with nearby outdoor environments.

To evaluate the indoor performance of the $NO_2$ and the OPC sensors, an experiment in an urban flat in central Beijing was performed during the non-heating season (May 2017). One PAM was deployed in the living area next
to two commercial instruments that were used to provide reference measurements: (1) a cavity attenuated phase shift spectroscopy instrument (CAPS Teledyne T500U) for $NO_2$ and (2) a portable commercial spectrometer (GRIMM 1.108) for particulate matter measurements (Table 2). During the experiment the occupants relied on natural ventilation adjusting the windows freely to achieve thermal comfort. Median indoor temperatures were 26.0°C (5%-95% range: 17.1°C – 28.8°C), and the median internal PAM temperature was 33.0 ºC (5%-95% range:
24.3°C-36.2°C), which is comparable with the temperature range during the non-heating season field deployment to participants (internal median temperature: 35.0°C, 5%-95% range: 28.5°C- 39.9°C, Figure A5).

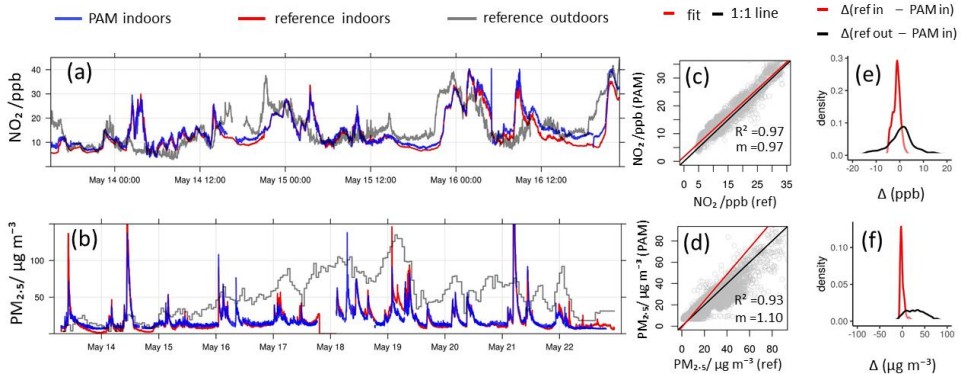

**Figure 4. Indoor co-location of a PAM with portable commercial instrumentation (Table 2) in an urban flat in China during the non-heating season. (a):** Time-series of $NO_2$ from the PAM (blue) and a cavity attenuated phase shift
spectroscopy (CAPS) instrument (red). The conversion of the raw measurements to ppb used the sensitivities extracted





using outdoor co-locations both during the heating and non-heating season (subsection 3.1) with the linear model (subsection 2.1). Outdoor NO₂ measurements (grey) were collected at PKU reference site (Table 2, 1-minute time resolution). Time resolution of measurements is 1-minute. **(b):** Time series of PM₂.₅ mass measured with the PAM (blue) next to a commercial portable spectrometer (GRIMM 1.108, red). Mass concentrations were calculated from particle counts within the size range 0.38 -17 μm and same aerosol density for both instruments. Outdoor PM₂.₅ mass measurements (grey) were collected at a nearby governmental station (Table 2, 1-hour time resolution). **(c) and (d):** Scatterplots show an excellent agreement between commercial instruments and miniaturised sensors making them suitable for the quantification of indoor pollution levels. The 1:1 line is in black and gradient m in red. **(e) and (f):** Density plots of the difference between measurements from the PAM and the indoor reference (red) are compared with the difference between the PAM and the outdoor reference (black).

The performance of the low-cost sensors in the indoor environment (Figure 4 and Figure A6) was comparable to the outdoor performance demonstrated in the previous section ($\overline{R^2}$ = 0.97, gradient m = 0.97, RMSE = 2 ppb for NO₂ (Figure 4c) and $\overline{R^2}$ = 0.93, gradient m = 1.10, RMSE = 7 μg m⁻³ for PM₂.₅ (Figure 4d)) proving their suitability for the quantification of indoor air pollution levels for these species.

Although this short experiment is only a "snap-shot" of indoor exposure, it shows that the measurement error of the PAM relative to established commercial instruments is negligible compared with the error in indoor exposure estimates introduced from using inadequate exposure metrics, in that case, outdoor measurements from the closest monitoring reference site. For example, using outdoor measurements from the closest monitoring station would have resulted in an over-prediction of indoor PM₂.₅ concentrations (moderated by attenuation effects of the building envelope) with an average difference of 30 μg m⁻³ (standard deviation: 29 μg m⁻³) which is significantly higher than the 7 μg m⁻³ RMSE value of the PAM (Figure 4f). While indoor NO₂ levels broadly followed outdoor levels, the range of the error in under-predicting and over-predicting exposure events is much broader (min-max range: -18 to 18 ppb; Figure 4e) compared with the error introduced from measurement uncertainties (-7 to 5 ppb). Such peak exposure events might be important triggers for acute health responses.

### 3.3 Performance of the PAM in non-static configurations

The aim of this section is to evaluate the PAM reproducibility and accuracy while in movement, with pedestrian and in-vehicle deployments.

### 3.3.1 Reproducibility of the PAM when not static

Multiple (in this case nine) PAMs were carried by a pedestrian while keeping an activity diary and walking between two indoor environments via a highly trafficked road in Cambridge, UK (weekday in January). Using NO measurements (the main traceable component from combustion engines) as an illustrative example, Figure 5a shows the simultaneous measurements of all PAMs as a time series and the scatterplots between the measurements of two of those PAMs separated into indoor (Figure 5b) and outdoor data (Figure 5c).

Significant changes of the pollution levels were observed when moving between the different environments illustrating the high granularity of personal exposure in daily life. Compared with the indoor environments, walking in traffic resulted in elevated pollution exposure events. As illustrated in the time series of Figure 5, the difference in pollution levels between the three micro-environments was significantly higher than the variability between PAM measurements.





Table 4 gives an overview of the correlations within the co-located moving network: In indoor environments an excellent agreement between all sensors (median $R^2 > 0.96$) was found, indicating a high sensor reproducibility. An exception was the $O_3$ sensor which showed poor between-sensor reproducibility due to very low indoor and outdoor concentrations (< 5 ppb) near the LOD of the sensor. The between-sensor correlations in the road

environment were lower than indoors (median $R^2 > 0.85$) due to highly heterogeneous air pollution concentrations driven by complex factors (e.g. canyon air mixing, moving vehicle sources, topology). This signifies that in such environments air pollution concentrations might differ on such a short spatial and temporal scale that even sensors that are less than one meter apart from each other capture a slightly different exposure profile.

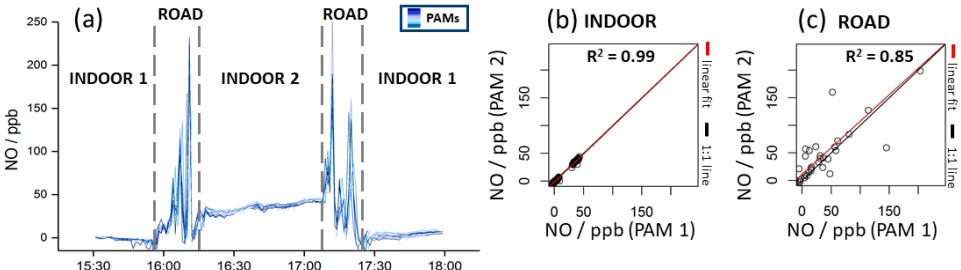

**Figure 5. Short-term deployment of nine PAMs carried simultaneously by a pedestrian moving between two indoor environments (laboratory, café) in Cambridge, UK, in January 2018. (a): Time-series of NO measurements from the PAM sensors (blue lines). (b) and (c): Scatterplots between two of those PAMs, whereby indoor data was separated from outdoor data. The 1:1 line in black; and linear fit line in red.**

     **Table 4. Correlations between PAM sensors: Adjusted $\overline{R}^2$ values of each sensor pair of the simultaneously carried**
**PAMs were determined. Median $R^2$ value of all combinations are presented in the table below. Very low $O_3$ levels (< 5 ppb) resulted in poor between-sensor correlations.**

|  | **Indoor** | **Outdoor** |
|---|---|---|
|  | median $\overline{R}^2$ | median $\overline{R}^2$ |
| **NO** | 0.99 | 0.87 |
| **NO₂** | 0.96 | 0.94 |
| **O₃** | *0.16* | *0.46* |
| **CO** | 0.99 | 0.95 |
| **PM₂.₅** | 0.99 | 0.85 |

When moving rapidly between different environments with different temperatures (i.e. from outdoors to a warmer indoor microenvironment) false peaks were observed in the EC sensor measurements (Figure A7) (Alphasense

Ltd, 2013). The response and recovery time following rapid temperature transitions was found to vary for different sensor types. To account for the false sensor responses, firstly an algorithm to identify those events was developed and then a 15-min window for CO and a 5-min window for NO, $NO_2$ and $O_3$ measurements was removed from the data (Figure A7 and Figure A8). Though it potentially excludes peak exposure events as rapid temperature changes often occur when people leave heated buildings and enter (colder) traffic environments to commute, this

correction method removes typically less than 0.1% of the exposure data set under daily life conditions. The PM measurements are not affected by these temperature transitions.





### 3.2 Accuracy of the PAM when not static

A PAM was mounted on the roof of a battery-powered vehicle equipped with multiple commercial instruments
(Table 2) mapping air pollution levels in London at speeds of up to 60 km/h for one day during the non-heating
season (Figure 6). Considering the high spatial variability of air pollution in traffic environments (see subsection
3.3.1), the accuracy of the PAM in a mobile configuration was high for all targeted pollutants ($\overline{R}^2 \geq 0.56$). To
illustrate the large degree of variability of air pollution concentrations over time, the investigated area was mapped
throughout the day multiple times with highest concentrations of $PM_{2.5}$ and $NO_2$ recorded during the morning rush
hour.

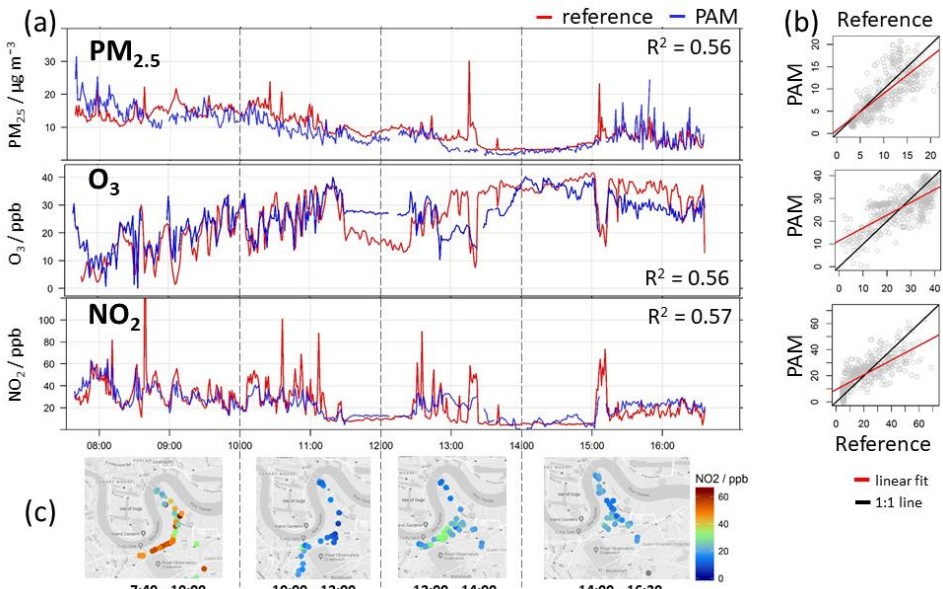

**Figure 6. The vehicle deployment in London, UK: a PAM was attached to a car equipped with multiple commercial instruments (Table 2) for four days. (a): Time-series of one-day measurements of the PAM (blue) and commercial instruments (red). (b): Corresponding scatterplots between measurements from commercial instruments and the PAM in motion in an urban environment. The 1:1 line in black; and linear fit line in red. (c): Maps (map data 2019 Google) of the mobile deployment over 2-hour windows illustrating the large temporal variability of $NO_2$.**

### 4 Discussion and conclusions

Mounting evidence points towards a causal link between exposure to air pollution and health outcomes. However,
due to current limitations in cost, maintenance and availability of instrumentation, most large-scale health studies
have focused on developed countries and have relied on low spatial and temporal resolution (generally outdoor)
air quality data as metrics of exposure, severely limiting causal inferences in epidemiological research worldwide.
Emerging low-cost sensing technologies can offer a potential paradigm shift in capturing personal exposure of the
population during daily life in addressing this critical shortcoming.

In this paper we demonstrated that, with suitable calibration and post-processing, the performance of currently
available low-cost air quality sensors, in this case incorporated into a highly portable personal monitor (the PAM)
is comparable with the performance of reference instrumentation across a wide range of conditions:



- in diverse outdoor environments (urban background and traffic);
- across seasons (over a wide temperature and RH range);
- in two geographical settings with differing air pollution levels and meteorological profiles (UK and China);
- in indoor environments (residential, laboratory, café) with varying emission sources, and
- static and in non-static deployments.

A critical important outcome of this study is that the performance of the sensors substantially exceeds that needed to quantify the differences between indoor and outdoor pollution levels, and thus to quantify exposure levels in a reliable manner.

There are certain performance caveats with the low-cost sensors used in this study, which once identified are likely to be addressed in future generations of sensors:

- The performances of the CO, NO and $NO_2$ sensors were found to degrade at temperatures above 40 $^0$C. In fact, such extreme environmental conditions were not encountered during the actual personal exposure
sample periods for which the PAMs were used, and the performance criteria discussed above were met.
- A limitation of all optical PM sensors, low-cost or reference, is that they cannot measure small particles below a critical size threshold (typically 200 – 400 nm). In this work we show that by appropriate local calibration, this shortcoming can be largely accounted for.

The toxicity of particles is also likely to depend on their chemical composition. Most national networks measure total mass only, and measuring particle chemical composition is currently largely the domain of the research community. A major challenge will be to develop techniques to allow routine PM composition measurements, both for the regulatory networks and for applications such as personal monitoring.

The key conclusion is that when suitably operated, highly portable air pollution personal monitors can deliver traceable high quality exposure metrics which can address scientific, health and policy questions for the indoor and outdoor environment in a way that has not been possible before. Mobile and static PAM networks have now been deployed in a range of health studies, and these will be the focus of future papers.

**Acknowledgments**

The authors would like to thank the following: Qiang Zhang and Kebin He (Tsinghua University) for PM outdoor data during the non-heating season, Paul Williams (University of Manchester) for providing the GRIMM instrument, Envirotechnology Services for reference mobile measurements. The authors would also like to thank MRC, Newton Fund, NERC and National Natural Science Foundation of China (NSFC Grant 81571130100) for funding for the AIRLESS project and MRC for funding of the COPE project.




**Author contribution**

**L.C. and A.K. have contributed equally to this manuscript. Conceptualisation:** L.C., A.K., B.B. and R.L.J.; **Sensor platform development:** L.C. and M.K.; **Sensor platform deployment:** L.C., A.K., Y.H., T.W., H.Z., Q.W. and S.F.; **Data curation:** L.C., A.K., O.A.M.P., A.D.A, M.K., Y.H., F.A.S., S.C. and B.B.; **Formal data**

5    **analysis:** L.C., A.K., O.A.M.P., A.D.A. and R.L.J.; **Resources:** M.H., J.K.Q., B.B., F.J.K., T.Z. and R.L.J.; **Software:** L.C., A.K. and M.K.; **Visualisation:** L.C. and A.K; **Writing original draft:** L.C. and A.K.; **Review and editing:** O.A.M.P., Y.H. and R.L.J.

*Competing interests.* The authors declare that they have no conflict of interest.

10    *Data availability.* All data can be provided by the corresponding author upon request.



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
