# Peer review of "Characterising low-cost sensors in highly portable platforms to quantify personal exposure in diverse environments."

_Atmospheric Measurement Techniques, 2019_

## Referee Comment (RC1) · Anonymous Referee #1 · 6 Jun 2019

Chatzidiakou et al describe calibration of a low-cost sensor package, called the PAM, in both indoor and outdoor environments in the UK and China. Overall the paper is written well, technically competent, and topically suitable for AMT.

My main concern is about the novelty of this manuscript. At this point there is a robust literature on the calibration of low-cost electrochemical and metal oxide gas sensors for use in outdoor environments. It is not immediately clear how this manuscript makes a significant contribution on that front.

Furthermore, this manuscript seems to miss many important citations when it comes to the use of low-cost gas sensors in outdoor environments. Here are several I can think

of quickly (this is certainly not an exhaustive list): -Malings et al, AMT, 2019 -Masson et al, Sensors, 2015 -Piedrahita et al, AMT, 2014 -Spinelle et al, Sensors and Actuators B, 2015 and 2017 -Zimmerman et al, AMT, 2018

In my opinion, the indoor evaluation is the most novel part of the manuscript. Most of the existing literature deals with outdoor evaluations of these sensors. Most of the published calibrations are empirical (e.g., the linear models used in this manuscript, machine learning calibrations, etc), so it is no guarantee that calibrations developed in outdoor co-locations will work well indoors. This manuscript shows that outdoor calibrations can transfer to indoor environments, which I think is a significant contribution. However, the section on the indoor deployment (section 3.2) is short and could be expanded.

Other comments: (1) Page 5 Lines 25-32 - I am a bit unclear on the corrections used for the OPC. Specifically I don't understand where the density comes into play. Once the raw particle mass is adjusted for RH, it seems like the regression against the TEOM should nominally account for all size and density effects.

(2) Page 6, Line 7 - the PAMs were given to study participants in the UK for two years. Were there pre- and post-calibrations to look for drifts? Two years is around the expected working lifetime for the alphasense ECs.

(3) Does Figure 2 show raw data or calibrated data?

(4) Table 3 - (a) why is the China - June column in italics? -(b) Are the $R^2$ and RMSE for the calibration data or the testing data? -(c) How sensitive are the calibration results to the selection of the calibration period? E.g., in Figure 3, 5 days are selected for calibration, and those 5 days seem to work well because concentrations changed significantly over the calibration period. If the calibration period was ~Jan 8-12, when concentrations were steadier (and lower), the calibration performance would presumably be poorer. How was the length of the calibration period determined?

---

## Referee Comment (RC2) · Anonymous Referee #2 · 11 Jun 2019

This manuscript presented results for their portable personal air quality monitor (PAM) measurements in outdoor, indoor and commuting microenvironments across seasons and in different geographic areas between UK and China. Overall the paper is well written with sufficient technical details, and in an area that of interest of AMT.

There are many sensor studies worldwide in recent years, which evaluated sensor performance with FRM/FEM methods, or intra-/inter-sensor variability, or in different microenvironments in real-world settings. The novelty of this manuscript by comparing and presenting co-locating sensor measurement results is not significant. Also by reading the literature review part, I feel like the manuscript missed some important citations such as from US EPA for sensor evaluation studies in ambient environments. However, this manuscript gets my attention when it says it is the first of a series of publications that aim to relate personal exposures to health outcomes. There are many sensor measurement studies, but seldom have been used to link with exposure and health studies. I would be interested to see more results for microenvironments other than outdoor. Results presented so far in this manuscript for indoor and commuting are limited.

The R2 from the study are typically higher than what I learnt from existing studies, which might partly related with the calibration method they used. I would also like to see the comparison, maybe in appendix, between results not calibrated by environmental conditions such as temperature and humidity with reference instruments because many studies deploy their sensor without calibration prior to testing. It would be good if the manuscript can show the inclusion of temperature/relative humidity and cross-sensitivity from other gas greatly improves the R2. Also, how the calibration period was selected? Does the calibration period in the range of typical concentrations for all pollutant measured, and in the typical range for temperature/relative humidity?

For the commuting environment, measurements were taken on the vehicle roof. It should be noted that when evaluating personal exposure, people are usually sitting inside of vehicle. There is also a difference between in-vehicle personal exposure and outside-vehicle ambient concentration. When developing future publications, I suggest authors to consider this when evaluating measurements for commuting microenvironment.

Specially, Page 2, line 20-23: suggest including more references Page 3, line 22-24: suggest including price for individual sensors as readers might be interested Page 9, table 3: why R2 is only 0.20 for NO2 and NO in Chine during non-heating season? The results do not seem to agree well with other results.

---

## Referee Comment (RC3) · Anonymous Referee #3 · 27 Jun 2019

Comments on Chatzidiakou et al. (submitted) Characterising low-cost sensors in highly portable platforms to quantify personal exposure in diverse environments

General comments: This manuscript describes an innovative and useful development for air pollution exposure assessment using novel miniature automated gas & particle sensor systems. The manuscript represents an important addition to the literature in this field on a topic that is of considerable international public health concern. Strengths of the study include the novelty and numbers of instruments evaluated; and the diversity of field evaluation environments studied.

Improvements could be made to the manuscript to address possible issues related

to the reproducibility of the science described. The results could also be presented in a way that clarifies the possible effects of arbitrary division of field data into test and training subsets (raised by earlier published review comments). Earlier review comments have also asked for clarification on key issue about extent of sensor drift. The manuscript would be strengthened by specifically addressing these points even if this is merely to provide adequate quantification of why arbitrary division of data, and sensor drift, are not important with the instruments evaluated in this research. I have suggested some possible ways of addressing these and other points below.

Specific comments: Abstract: 1. It would appear beneficial to include brief selected quantitative summary information on the precision, accuracy and lack of (or extent of) measurement drift over time for the instruments evaluated.

Introduction: 2. It is indicated that 'this paper further aims to create a roadmap for calibration and validation of portable monitors suitable for personal exposure quantification'. I'm not sure what you meant by this, and wondered if this aim could be stated in a clearer way?

Methods: 3. On the general scientific point about [in an ideal world] research being reproducible by other researchers to accelerate discovery (Munafò et al., 2017) it would be helpful to clarify if the monitoring systems being tested are [or will be] commercially available to wider research communities beyond consortium of authors. I appreciate that the sensors are specified, but also there may be a substantial number of person-years invested in the design and construction of the monitoring systems that it may be appropriate to elaborate on. I also appreciate it is fine to report on instruments prior to commercialisation as the ideally reproducible science specified by Munafò et al. may not always be practical in the short to medium term.

4. It would be helpful to expand on the principle of operation of the gas sensors used. The references given are useful but these tend to focus on 3-electrode systems rather than the innovative 4-electrode system that you have used. I think you could make more

of this by explaining the principle of operation of this system in a more expanded way that is accessible to a wider audience beyond specialists in the construction of electrochemical sensors e.g. a diagram similar to diagrams of 3-electrode systems in earlier references would appear to be helpful. Stetter & Li 2008 [referred to in manuscript] indicate how gas membranes, electrolytes and electrodes are key components of amperometric gas sensors. I wondered if the manuscript would be improved by providing information on the characteristics of these sensor components for the systems tested? How does gas reach gas sensors e.g. diffusion or active fan etc?

5. You refer to earlier work by Mead et al. 2013 in relation to sensor linearity and quantified limit of detection (LoD). This earlier paper seems to be focused on 3-electrode sensors. Is it ok to directly translate the finding to the newer 4-electrode sensor, or is separate quantification of the linearity and LoD of the newer sensor also required?

6. A strength of your paper is that you use a conceptually simple calibration model (Equation 1) cf. potential over-fitting of calibration data in other sensor evaluation papers. To emphasise this strength I think it would be beneficial for you to present the results of the calibration in terms of the equations fitted. If the information on these equations is too extensive they could be included in Supplementary Information.

7. It would appear useful to make your field calibration data available as a published dataset linked to your paper to allow other researchers evaluating different, but similar types of, electrochemical sensors to compare similar calibration data to your results.

8. Perhaps it would be beneficial to add a reference for Mie scattering on p5.

Results:

9. In Fig 3 why did you decide to 'draw the line' at the specific point in the time series between panel (a) and panel (b). Was this splitting of the data done in an entirely objective manner, or did you optimise the split to get 'good' calibration results e.g. maximising R2 and/or other metrics of agreement between sensor and reference data?

From what I can see your results look as if they may be fairly robust to different selection possibilities, but I think (as suggested by the earlier reviewers) it would be useful to make this clearer in the manuscript, and that you could strengthen the manuscript by quantifying the effect of any arbitrary splitting of the data against other arbitrary (or ideally - objective) data splitting decision possibilities. If you were able to develop a robust objective way of doing this it would help with reproducibility concerns outlined above.

10. The preceding point about split between training and test could similarly be clarified in more detail for data in Figs 4, 8, A2, A3, A4 and A6 (currently it is not clear what calibration data has been used to adjust the sensor outputs in these figures).

11. The quantification of sensor drift over the duration of measurements [and beyond duration of field measurements as you suggest in response to earlier reviewers] could also be beneficially included in the manuscript in the vicinity (perhaps in Figure captions) of Figures 4, 8, A2, A3, A4 and A6.

12. I recommend giving dates and the number of hours of measurements for each of the deployments in the columns in Table 3.

13. In Table 3, instead of giving RMSE as a percentage of maximum (which may [by definition] be an outlying point) would it not be better to specify the mean reference concentration and specify the RMSE as a percentage of that mean?

14. On p10 you indicate that an RMSE of less than 16% of maximum concentration is negligible. I think it may be better to express as percentage of mean (i.e. preceding point) and let your readers reach their own description of the level of agreement.

15. On page 11 second paragraph were the extreme temperatures you refer to recorded inside the monitor enclosure and within the sensors?

16. Do you know why the O3 sensor appeared to be less affected by the extreme temperatures cf. the CO, NO and NO2 sensors?

17. Your reference to the participants on p11 could be given with further detail e.g. geographical location / time of year etc. to provide context to the comparison with the field evaluation measurements.

18. In Figure 4 caption it would be helpful to specify how far the 'PKU reference site' and the 'nearby government monitoring site' were from the indoor measurement location.

19. On p12 you assert that your results prove the suitability of the low cost sensors for quantification of indoor air pollution. It would seem appropriate to qualify the promising indication of suitability over timescales and conditions similar to the calibration period/conditions?

20. Do you have information on the response times of the sensors, and could any difference in response time between sensors for different pollutants affect the cross sensitivity calibrations determined at the static monitoring locations?

21. When the PAM was mounted on the roof of the vehicle how were the sensor inlets orientated in relation to airflow? Did the varying speed of airflow have any effect on agreement between sensors and reference instruments?

22. In Fig 6 where, and what sort of, measurements were being made between: 11.30-12.30 e.g. was vehicle static or in a quiet road with no other vehicles? Do you know why O3 measured by sensor and reference instruments diverges during this time period?

Reference: Munafò, M.R., Nosek, B.A., Bishop, D.V.M., Button, K.S., Chambers, C.D., Percie du Sert, N., Simonsohn, U., Wagenmakers, E.-J., Ware, J.J., Ioannidis, J.P.A., 2017. A manifesto for reproducible science. Nature Human Behaviour 1, 0021.
* * *

---

## Author Response (AR1)

**Interactive comment* on "Characterising low-cost sensors in highly portable platforms to quantify personal exposure in diverse environments" by Lia Chatzidiakou et al.**

**Response to anonymous referee #3**

Overall response: We would like to thank reviewer #3 for their useful and insightful comments. The reviewer recognises the importance of the manuscript and asks for technical clarifications. However, we would like to emphasize that this manuscript is not aiming to provide a *generic* evaluation of all portable air quality sensor types but instead aims to characterise the performance of this specific configuration with currently available sensor variants for specific environmental conditions, as a necessary precursor to a series of papers on exposure and health impacts using these PAMs. While the various methodologies we used would be of value to the scientific community, as the field of air quality monitoring evolves, different sensor types, models and configurations might be expected to lead to different results.

**Detailed response**

**Abstract 1:** *"It would appear beneficial to include brief selected quantitative summary information on the precision, accuracy...."*

Response (added): ....Overall, the air pollution sensors showed high reproducibility ( $R^2$ : 0.80 -1.00, mean  $R^2 = 0.93$ ) and excellent agreement with standard instrumentation ( $R^2$ : 0.56 - 0.99, mean  $R^2 = 0.82$ ) in outdoor, indoor and commuting microenvironments across seasons and different geographical settings.

**Introduction 2:** "this paper further aims to create a roadmap for calibration and validation of portable monitors suitable for personal exposure quantification.... wondered if this aim could be stated in a clearer way?"

Response (re-phrased as follows pg 2 ln 35-37): As the field of novel air pollution sensing technologies expands rapidly, this paper further aims to provide methodological guidance to researchers from diverse disciplines on how to comprehensively calibrate and validate portable monitors suitable for personal exposure quantification.

**Methods 3**. "*it would be helpful to clarify if the monitoring systems being tested are [or will be] commercially available to wider research communities …. also there may be a substantial number of person years invested in the design and construction of the monitoring systems…"*

Response (added as follows pg 2 ln 39-41): The PAM has been developed at the Department of Chemistry, University of Cambridge in collaboration with Atmospheric Sensors Ltd. It is now commercially available (independently from the University of Cambridge) from Atmospheric Sensors Ltd (Model AS520, http://www.atmosphericsensors.com).

**Methods 4 and 5.** "It would be helpful to expand on the principle of operation of the gas sensors used... accessible to a wider audience... e.g. a diagram similar to diagrams of 3-electrode systems in earlier references" and on the same issue of 4-electrode EC sensors "Is it ok to directly translate the finding to the newer 4-electrode sensor, or is separate quantification of the linearity and LoD of the newer sensor also required?"

Response: (added in paper pg4 In1-10): These EC sensors are the A4 variant from Alphasense (ref) and operate on a four electrode system. The principle of operation of the four electrode system is identical to that of the earlier variants of three-electrode system with an additional electrode, the auxiliary (or non-sensing) electrode to compensate for the temperature dependence of the cell potential. .... The linearity and LoD of the 4-electrode sensors (when integrated in the PAM) have been tested under laboratory conditions following the same methodology as described in Mead et al. 2013, yielding very similar results.

Response to referee: Rather than include inevitably incomplete sensor details (as much is proprietary to the manufacturer), we feel it more appropriate -to refer readers (and the

referee) to the manufacturer's website for further details about the principles of operation of the individual sensors integrated in the PAM. The 4th electrode was implemented to correct for systematic errors and not to change the LOD. Hence, we expect the linearity and LOD to be comparable with those of the 3-electrode sensors.

**Methodology 6.** "A strength of your model is that you use a conceptually simple model.....present results of the equations fitted... in Supplementary Information".

Response to referee: Such calibration factors are likely to be location specific. Making them public, would risk encouraging other researchers taking them as generic.

Methodology 7. "It would appear useful to make your field calibration data available" Response: Data availability: https://doi.org/10.17863/CAM.41918

**Methodology 8.** "*Add references for Mie*" Response (pg5 In 7-8, references added):**

We have included a suitable reference.

Mie, G. Beiträge zur Optik trüber Medien, speziell kolloidaler Metallösungen. Ann. Phys. 1908, 330, 377–445. Bohren, C. F.; Huffman, D. R. Absorption and scattering of light by small particles; Wiley, 1988.

**Methodology 9 &10.** "robust objective way of splitting the data to calibration-validation periods".... "Currently it is not clear what calibration data has been used to adjust the sensor outputs in these figures"

Response (added to the paper):

To evaluate the performance of the linear model, the datasets were split into training (i.e. calibration) and validation periods to first extract the calibration parameters and then apply them to the validation set and compare the measurements with those from reference instruments (referred to as "calibration - validation" method). The training sets ranged from 1 to 16 days, and the adjusted coefficient of determination ( $\overline{R}^2$ ) remained stable for training periods longer than 3 days. Therefore, approximately a third of the dataset was selected as a training set. As relationships in these linear models should ideally not be extrapolated beyond the range of the observations (including meteorological conditions), the calibration periods et al., 2017). Once the performance of the model was established in diverse environments, we used the full co-location periods to determine the agreement between PAM sensors and reference instruments.

Response to referee (not in the paper):

Calibration of gaseous sensors for the China deployment: The China deployment lasted about one year, and no evidence of a significant drift of the gaseous sensors was found. Using a unified **training** dataset from the outdoor co-locations **both in the heating and nonheating season** the coefficients of the linear model were determined. As the variation between seasons was greater than the variation within seasons and covered a wide range of environmental conditions and pollution levels, the choice of the training period did not affect the performance of the model. The coefficients determined in the training set were then applied to the **validation** periods of both seasons and were also used to convert raw measurements to ppb in the indoor deployment. In this way, we prove that the calibration parameters determined in outdoor co-locations are suitable across a range of environmental conditions and for indoor deployments too, and therefore for personal exposure as people spend as much as 90% of their time in indoor environments.

Calibration for the UK deployment: The PAMs were co-located outdoors next to reference instruments in 2015, and were then deployed for two years continuously to participants of the COPE project. The PAMs were again co-located next to reference instruments in 2017 after the completion of the project. The change in the sensor sensitivities

did not show a systematic error. Individual sensor drifts were quantified and were corrected by linearly interpolating sensitivities between the start and the end of the deployment.

| Response: Added to Table 3 first column |              |         |            |            |            |  |
|-----------------------------------------|--------------|---------|------------|------------|------------|--|
|                                         | Coloc        | ation   | Start      | End        | Total time |  |
|                                         | China        | (Dec-   | 28/12/2016 | 15/01/2017 | 447 hours  |  |
| Jan)                                    |              | ·       |            |            |            |  |
|                                         | UK           | (Oct-   | 27/10/2017 | 13/11/2017 | 408 hours  |  |
| Nov)                                    |              | -       |            |            |            |  |
|                                         | China (June) |         | 28/06/2017 | 16/07/2017 | 432 hours  |  |
|                                         | UK           | (April- | 26/03/2018 | 10/04/2018 | 342 hours  |  |
| May)                                    |              |         |            |            |            |  |

Results 12. "Dates and measurement times in Table 3"

**Results 13 & 14. "Express RMSE as percentage of mean reference" -**

Response: We have added mean values of each pollutant during the co-location periods. We report RMSE as a percentage of the maximum (as the 95% percentile), as peak exposure events are more relevant in health studies. The reader is now also able to calculate the RMSE as percentage of the mean reference values.

**Results 15.** pg 11 ln 6 "were the extreme temperatures recorded inside the monitor enclosure and within the sensors?"

Response to referee: The temperature inside the PAM is on average 7°C higher than ambient temperatures. The monitors were co-located in metal shelters on the roof of a research building in the Peking University campus (Figure below), and were exposed to direct sun further increasing the temperature inside the PAM.

Figure (not included in manuscript): Co-location of 60 PAMs at PKU during the heating and non-heating season for a four-week period. PAMs were placed in custom-made protective shelters in proximity to the inlets of reference instruments and were connected on mains. Response added: In Table 3 added "internal conditions of the PAM"

**Results 16.** "Why was  $O_3$  sensor less affected by heat then  $NO_x$  and  $CO_2^2 - O_3^2$  concentrations were higher than usual?"

Response to referee: The principles of operation of the NO2 and O3 sensors are very similar. It is possible that the O3 sensors perform better because (a) the concentrations of O3 were higher in the summer and (b) the sensor is further away from the internal battery of the PAM and therefore slightly cooler than the other sensors. The next generation of PAM hardware will integrate an insulation layer between battery and sensors. NO was close to the LOD (maximum= 5ppb, mean =1 ppb).

**Results 17.** "Your reference to the participants on p11 could be given with further detail e.g. geographical location / time of year etc. to provide context to the comparison with the field evaluation measurements."

Response: FIGURE A5 CAPTION Added: The co-location with the reference instruments on the roof of Peking university took place from 28/06/2017 to 16/07/2017, the field deployment was conducted in Beijing and Pinggu from 22/05/2017 to 26/06/2017.

**Results18.** *"In Figure 4 caption it would be helpful to specify how far the 'PKU reference site' and the 'nearby government monitoring site' were from the indoor measurement location"*

Response ADD TO FIG 4 CAPTION: PKU roof is 5.3 km and the governmental reference station (Haidianwanliu) is 6 km away from the location of the indoor experiment.

**Results 19.** "On p12 you assert that your results prove the suitability of the low cost sensors for quantification of indoor air pollution. It would seem appropriate to qualify the promising indication of suitability over timescales and conditions similar to the calibration period/conditions?"

Response: Add to PAGE 12 LINE 34:

The conversion of the raw measurements to ppb used the sensitivities extracted using outdoor co-locations both during the heating and non-heating season (subsection 3.1) with the linear model (subsection 2.1).

....proving the suitability of this monitoring platform to quantify indoor air pollution levels for these species provided they have been adequately calibrated in the local environment.

Results 20. "Do you have information on the response times of the sensors..."

Response to referee: Response times have been determined in the Alphasense Technical Specification sheets of each sensor (see references in Section 2.1, page 3) and ranged between  $t_{90} < 25s$  for CO (from 0 to 10ppm) and  $t_{90} < 60s$  for O3 (from 0 to 1ppm). As the response times are smaller than the time resolution for the PAM measurements we do not expect them to affect the way of correcting the sensors' cross sensitivities.

**Results 21.** "When the PAM was mounted on the roof of the vehicle how were the sensor inlets orientated in relation to airflow? Did the varying speed of airflow have any effect on agreement between sensors and reference instruments?"

Response (added to manuscript pg 15): The PAM was mounted on the roof with the OPC inlet facing forwards and the EC sensors facing to the sides (see photo below). The reference instrument inlets were located on the car roof as well. The speed mostly varied between 5 and 20 km/h (see histogram below). There was no correlation between car speed and RMSE error in the gaseous and particulate measurements. The OPC contains an airflow measurement unit which compensates for any wind or internal flow dependence.

The histogram below shows the distribution of the speed measurements covering the same time period as Figure 6 in Section 3.2.

**Results 22.** *"11.30- 12.30 e.g. was vehicle static Do you know why O3 measured by sensor and reference instruments diverges during this time period?"*

Thank you for bringing this to our attention. During this period the vehicle was stationary, and we have therefore removed this data.

**Response to anonymous referee #2**

**Overall response:** Thank you for taking the time to provide useful comments. Referee #2 mentions that this manuscript is essential to underpin the validity of personal exposure measurements collected in two major health studies (APHH-Beijing, Theme3: AIRLESS https://doi.org/10.5194/acp-19-7519-2019 and COPE study <a href="https://doi.org/10.5194/acp-19-7519-2019">https://doi.org/10.5194/acp-19-7519-2019</a> and COPE study <a href="https://doi.org/10.5194/acp-19-7519-2019">https://doi.10.1136/bmjopen-2016-011330</a>). The specific aim of the manuscript is therefore to evaluate a specific sensor package (the PAM) with a comprehensive, robust and reproducible methodology, rather than individual sensors or a generic sensor package.

The referee states that "there are many sensor studies worldwide in recent years", feeling this work is not novel. However, there are concerns remaining in the scientific community regarding the validity of measurements collected with miniaturised portable sensors. For example, **a recent literature review on portable sensors** (Thompson, 2016 https://doi.org/10.1016/j.teac.2016.06.001) states that "current technology for inexpensive portable sensors is not sensitive or specific enough to meet demands" while a commentary article in Nature 2016 (https://www.nature.com/news/validate-personal-air-pollution-sensors-1.20195) disregards novel technologies due to "questionable air quality data". Such opinions act as a barrier in adopting innovative methods that could revolutionise multiple disciplines including epidemiological research and the built environment and have significant societal benefits. Extending beyond the specific aim outlined above, we feel that this manuscript does also contribute significantly and positively to the wider literature of novel portable sensor technologies.

Detailed response: Temperature and relative humidity were not included explicitly in the linear model for the calibration of the electrochemical sensors. The effect of relative humidity on particulate matter estimations has been quantified in a previous publication (Di Antonio, A., et al., 2018. *Sensors*, https://doi.org/10.3390/s18092790). The cross-interference of other gases on the electrochemical sensors is covered in the manufacturers specifications and is beyond the scope of this work.

Calibration periods were selected based on campaign time periods not conflicting with deployments of the PAM to participants. The training set was about 1/3 of the total observations, an arbitrary choice. We used a combined training set from the winter and the summer co-locations. In that way, the selected training periods of each season become less important as the variation in pollutant levels between seasons is much greater providing the necessary wide range of calibration conditions.

The vehicle deployment aimed to evaluate the performance of the PAM in movement and did not aim to capture personal exposure of an individual within a vehicle. Forthcoming publications focus on the magnitude and duration of personal exposure in diverse microenvironments (including indoor locations and different commuting modes) during daily life activities.

The selected references on static outdoor co-locations are inevitably selective, and are not exhaustive of the large body of evidence on novel technologies. However, there is a lack of publications on the performance of portable platforms in diverse microenvironments, as presented in this manuscript.

Prices for individual sensors can be provided by the manufacturers. Low  $R^2$  values especially for the NO and NO2 sensors were noticed at temperatures above 40 C (non-heating season in Beijing), which is above operational specifications and were not recorded during the participant deployment.

**Response to anonymous referee #1**

Overall response: Thank you for taking the time to provide valuable feedback. The aim of this paper was to validate the performance of a specific novel personal air pollution monitor (PAM) when capturing personal exposure. While outdoor co-locations next to certified instruments have been widely adopted by researchers and governmental organisations to validate the performance of sensors in the field, this paper goes beyond those current guidelines by validating the PAM in indoor and commuting microenvironments, and thus demonstrating that novel sensing technologies can provide reliable personal exposure measurements.

The importance of this work is two-fold:

- Addressing concerns which remain in the scientific community regarding the suitability of novel sensing technologies for policy purposes and health studies. Such opinions act as a barrier in adopting innovative methods that could have significant societal benefits. In that sense,
- 2) This paper is the first of a series of publications that, together with detailed medical outcome determinations, aim to identify underlying mechanisms of specific air pollutants on health, and is necessary to validate in the open literature the performance of the PAM. Forthcoming publications will also focus on the modification effects of the indoor environment on personal exposure.

**Detailed comments:**

(1). OPC corrections: The detailed RH correction algorithm can be found in: Di Antiono, A., et al., 2018. *Sensors*, https://doi.org/10.3390/s18092790. A constant density was assumed for both, the reference instrument (Fidas Palas 200S, see Table 2, pg 7) and the portable instrument. It is true that in general, scaling to the reference compensates for density effects.

(2). Sensitivity drifts of sensors: The PAMs were co-located at the beginning and at the end of fieldwork with the reference instruments at the Department of Chemistry, UCAM. The change in sensitivities of the gaseous sensors was less than 10% and was therefore not included in the manuscript. The topic has been covered in a previous publication of the group that found similar drifts over an 11-month period (Mead, M.I., 2013, *Atmospheric Environment*, https://doi.org/10.1016/J.ATMOSENV.2012.11.060).

(3). Figure 2: This figure is an illustrative example of the methodology used to validate the performance of linear models used to convert raw units to physical measurements. The Figure presents calibrated data.

(4) (a) Chinese deployment during the non-heating season: Results are presented in italics due to the exposure of the sensors to very high temperatures. However, such temperatures were not encountered during the deployment to participants. (b) RMSE and adjusted R^2 for the combined training and testing period. (c) We used a combined training set from the winter and the summer co-locations. The training set was about 1/3 of the total observations. The sensitivities from the outdoor co-locations were then used to calibrate the indoor measurements: This proves that provided there is a diverse enough training set (both in terms of temperature and pollution levels) the linear model performs adequately in different conditions. In that way, the selected training period of each season becomes less important as the variation between seasons is much greater providing the necessary wide range of calibration conditions.

**Characterising low-cost sensors in highly portable platforms to quantify personal exposure in diverse environments.**

Lia Chatzidiakou1, Anika Krause1, Olalekan A.M. Popoola1, Andrea Di Antonio1, Mike Kellaway2, Yiqun Han3,4,5, Freya A. Squires6, Teng Wang4,7, Hanbin Zhang3,5,8, Qi Wang4,7, Yunfei Fan4,7, Shiyi Chen4, Min Hu4,7, Jennifer K. Quint9, Benjamin Barratt3,5,8, Frank J. Kelly3,5,8, Tong Zhu4,7, Roderic L. Jones1

1Department of Chemistry, University of Cambridge, Cambridge, CB2 1EW, UK 2Atmospheric Sensors Ltd, Bedfordshire, SG19 3SH, UK

3 MRC-PHE Centre for Environment & Health, Imperial College London and King's College London, London, W2 1PG, UK
 4 College of Environmental Sciences and Engineering, Peking University, Beijing, 100871, China
 5 Department of Analytical, Environmental and Forensic Sciences, King's College London, London, SE1 9NH, UK

6 Department of Chemistry, University of York, York, YO10 5DD, UK 7The Beijing Innovation Center for Engineering Science and Advanced Technology, Peking University, Beijing, 100871, China.

[revised manuscript text omitted]

**Method Sampling Interval Parameter Spatial coordinates Global Positioning System (GPS) 20 sec Background noise Microphone 100 Hz 100 Hz Physical activity Tri-axial accelerometer Temperature Band-gap IC 4 sec Relative Humidity (RH) Capacitive 4 sec PM1, PM2.5, PM10 Optical Particle Counter (OPC) 20 sec CO, NO, NO2, O3 Electrochemical sensors (EC) 100 Hz**

User-friendly, bespoke software (Supplementary materialMaterial Figure A1) has been developed to automate the management and post-processing of the large volume of raw data collected with the PAM network. Data is held in a PostgreSQL relational database management system which has an unlimited row-storage capacity and allows the querying of large quantities of data

in a flexible manner while maintaining performance as the volume of data grows. Post-processing was performed in R software (R Development Core Team, 2008) (Supplementary material Material Figure A1) following the methodology outlined in this paper.

**2.1 Measurements of CO, NO, NO2 and O3**

- 5 The principle of operation of all commercially available miniaturised gaseous sensors currently involves measuring changes in specific properties of a sensing material (e.g. electrical conductivity, capacitance, mass, optical absorption) when exposed to a gas species (Morawska et al., 2018). The PAM integrates small (20mm diameter) electrochemical (EC) sensors based on an amperometric principle of operation (Stetter & Li, 2008) for the quantification of carbon monoxide (CO), nitric oxide (NO), nitrogen dioxide (NO2) and ozone (O3). These EC sensors are the A4 variant from Alphasense (NO-A4 (Alphasense Ltd,
- 10 2016a), CO-A4 (Alphasense Ltd, 2017a), NO2-A43F (Alphasense Ltd, 2016b), Ox-A431 (Alphasense Ltd, 2017b)) and operate on a four electrode system. The principle of operation of the four electrode system is identical to that of the earlier variants of three-electrode system (Alphasense, 2013) where the conventional setup of working electrode, counter electrode and reference electrode is supplemented with an additional electrode, the auxiliary (or non-sensing) electrode to compensate for the temperature dependence of the cell potential (Popoola, Stewart, Mead, & Jones, 2016). Earlier variants of EC sensors
- 15 used in this paper have been extensively characterised in laboratory conditions and in static outdoor dense sensor networks (Mead et al., 2013). Those studies provided evidence that, after appropriate post-processing, the sensors had a linear response to the targeted pollutants and achieved excellent performance with limits of detection (LOD) < 4ppb demonstrating their suitability for atmospheric air quality measurements. The linearity and LOD of the 4-electrode sensors (when integrated in the PAM) have been tested under laboratory conditions following the same methodology as described in Mead et al. 2013, yielding very similar results.
- Currently, standards for the calibration and performance evaluation of EC sensors focus on industrial applications (British Standards Institution, 2017). Following those standards, a widely adopted approach to calibrate EC sensors are gas chamber experiments to determine offset (baseline) and sensitivity (gain). To address the lack of standards for novel sensing technologies, a number of researchers and governmental organisations are developing protocols and guidelines to evaluate sensor/monitor performance in the laboratory and in the field, such as the European Metrology Research Programme of EURAMET (Spinelle, Aleixandre, Gerboles, & European Commission. Joint Research Centre. Institute for Environment and Sustainability., 2013), the European Standardisation Committee (CEN/TC 264/WG 42, 2018) and US-based groups (Long, Beaver, & Williams, 2014) (AQ-Spec, 2017).
- 30

Building on those protocols, the EC sensors were calibrated by co-location with certified reference instruments in similar environmental conditions and same geographical area where the monitors had been or were to be deployed. The considerable advantage of this approach over laboratory calibration includes the exposure of the sensor to the actual air pollution and temperature/relative humidity conditions under which it is expected to operate, as well as the assessment of any site-specific potential cross-interferences. A linear regression model (Equation 1) was applied to the co-location data to determine the calibration parameters used to convert raw sensor signals (mV) to mixing ratios (ppb). Temperature effects were corrected through the auxiliary electrode AE which might have a different sensitivity to the working electrode WE ( $a \neq b$ ). The crosssensitivities between the NO2 and O3 measurements were corrected via parameter *c* (the cross sensitive gas *Y* is NO2 for O3 measurements and vice versa). As the CO and NO sensors were found to be sufficiently selective, *c* was set to zero for the calibration of those sensors.  $\left[X\right]_{ref} \ = \ a \ WE_X + \ b \ AE_X + \ c \ WE_Y + d$

reference measurement of pollutant X [ppb] sensitivity of the working electrode [ppb / mV]

periods to determine the agreement between PAM sensors and reference instruments.

raw signal of the working and auxiliary electrode [mV]

where [X]ref

5

10

30

a $WE_X$  ,  $AE_X$

b

*c* cross sensitivity with gas Y [ppb / mV], *c* = 0 for CO and NO *WEY* raw signal of the working electrode of the cross sensitive gas Y [mV] *d* intercept [ppb]
To evaluate the performance of the linear model, the datasets were split into training (i.e. calibration) and validation periods to first extract the calibration parameters and then apply them to the validation set and compare the measurements with those from reference instruments (referred to as "calibration - validation" method). As relationships in these The training sets ranged from 1 to 16 days, and the adjusted coefficient of determination (*R*2) remained stable for training periods longer than 3 days. Therefore, approximately a third of the dataset was selected as a training set. As relationships in these linear models should ideally not be extrapolated beyond the range of the observations (including meteorological conditions), the calibration periods were made sufficiently long to covercovered the temperature and concentration ranges in which the sensors were deployed

(Cross et al., 2017). Once the performance of the model was established in diverse environments, we used the full co-location

sensitivity of the auxiliary electrode [ppb / mV] (accounts for temperature)

**2.2 Particulate mass measurements**

[revised manuscript text omitted]